# The Sport Training Process of Para-Athletes: A Systematic Review

**DOI:** 10.3390/ijerph19127242

**Published:** 2022-06-13

**Authors:** Manuel Rodríguez Macías, Francisco Javier Giménez Fuentes-Guerra, Manuel Tomás Abad Robles

**Affiliations:** Faculty of Education, Psychology and Sport Sciences, University of Huelva, 21071 Huelva, Spain; manuvva6@gmail.com (M.R.M.); manuel.abad@dempc.uhu.es (M.T.A.R.)

**Keywords:** parasport, para-athletes, adaptive sports, impairment, disability

## Abstract

The aim of this systematic review was to identify the main factors affecting the training process of para-athletes, as well as the barriers they encounter. For this purpose, a systematic review was carried out in accordance with the PRISMA declaration guidelines, in which six databases were analysed (Web of Science, Scopus, SportDiscus, Pubmed, Eric, and PsycInfo). A total of 19 articles were selected for analysis after applying the inclusion criteria. The results show that the figures of the coach and families in the sporting and social contexts, respectively, had a relevant influence on the training process of para-athletes. Furthermore, in terms of psychological aspects, stress reduction, the importance of self-esteem, and motivation were highlighted. On the other hand, there are some barriers hindering the training and performance of athletes, which are related to the lack of financial support, lack of visibility in the media, and dependence on other people. These considerations can be of great help to coaches and competent institutions in the field (Paralympic committees, federations, etc.) in order to improve the training process and performance of para-athletes and to eliminate the barriers encountered by this group, promoting policies which facilitate access to sports for people with disabilities.

## 1. Introduction

In their training process, athletes learn and develop a series of physical, technical, tactical, and psychological skills which are adapted to their biological and psychological characteristics [1]. Therefore, the training process can be defined as an uninterrupted period parallel to the evolutionary development of the athlete, in which there is not only the influence of motor aspects but also factors related to cognitive and affective-social processes [2]. When analysing the most important factors, Kidman [3] considered that parents are very powerful socialising agents, as they can exert a great influence on their children in relation to the sport in which they specialise, as well as their adherence to it. On the other hand, Martinent and Decret [4] stated that the sports training process and performance are conditioned by psychological aspects and physical fitness. According to Mujika [5], psychological aspects are determinant when facing a training session or a competition, while Sosa et al. [6] and Williams and Krane [7] believe that self-confidence, motivation, concentration, and the control of stress and anxiety are of vital importance, both when it comes to achieving the proposed objective and in order to reach maximum performance.

In the case of athletes with disabilities, there is also previous research which indicates the existence of numerous factors influencing their training. Thus, Willis et al. [8] observed that the social context has a relevant influence on this process, while Williamson et al. [9] argued that the most important factor during the training of athletes with a disability is the coach’s knowledge of aspects related to rehabilitation, skill level, awareness, and knowing how to provide the athlete with constant safety. Additionally, Durstine et al. [10] emphasised that for good sport training of people with disabilities, programmes should focus on flexibility, balance, accessibility, safety, enjoyment, cardiovascular endurance, agility, and muscular strength. Nevertheless, athletes with a disability often have to overcome environmental, structural, social, medical, and economic barriers in order to train [11], which hinders their training and participation in sports [12]. DePauw and Gavron [13], Rimmer et al. [14], and Shields et al. [15] also considered that there is a number of physical, emotional, and psychological barriers influencing the training and participation of these athletes. Sobiecka et al. [16] found that one of the constraints for sport organisations not providing good training for para-athletes is the lack of funding and that, in sport clubs, the training process is not subject to any coordination.

Thus far, numerous studies have provided extensive knowledge on the training and coaching process of non-disabled people, although it is unclear whether these findings are transferable to para-athletes, as research in this area is at a nascent stage [17]. The literature draws attention to the lack of sufficient empirical research on the important components of training para-athletes at the highest levels of sporting prowess and the obstacles they encounter [16]. There is, therefore, little understanding of the mechanisms and processes by which athletes with a disability are trained to participate more or less effectively in competition-oriented physical activity [8]. In this regard, it should be borne in mind that both influences and constraints may depend on the type of sport in question, which could be the focus of future research. Thus, further research is needed on the factors influencing the sport training of those athletes who aspire to compete at the elite level of a given sport [18], as well as studies analysing the importance of these factors throughout the training process. In addition, in order to facilitate the acquisition of an optimal competitive level [19], it is convenient to highlight the need for adequate planning of these factors, as well as detection of the barriers affecting the training and participation of athletes with disabilities in sports.

For this reason, in order to increase knowledge about the factors influencing the training process of para-athletes, as well as to identify the barriers encountered by these athletes in this process, a systematic review was carried out. The research questions were the following: (1) What factors have the greatest influence on the training process of para-athletes, and (2) what are the barriers encountered throughout this process? Thus, the aim of this systematic review was to determine the factors influencing the training process of para-athletes, as well as the barriers they encounter.

## 2. Materials and Methods

In order to undertake this systematic review, the following method was used: the Preferred Reporting Items of Systematic Reviews and Meta-Analyses (PRISMA) statement and the practical guide for systematic reviews [20,21,22].

### 2.1. Eligibility Criteria

The inclusion criteria used for the selection of manuscripts were as follows: (1) full-text articles, (2) the subjects had to be Paralympic athletes, and (3) manuscripts had to be written in English, Spanish, or Portuguese. On the other hand, the exclusion criteria were (1) studies not related to the training process of Paralympic athletes, (2) systematic reviews or literature reviews, and (3) theses, book chapters, or conference proceedings.

Papers meeting all inclusion criteria were incorporated into the systematic review. In order to reduce selection bias, each study was reviewed independently by two authors who established whether or not the manuscripts met the inclusion criteria. In case of discrepancies, these were resolved by the third researcher.

### 2.2. Search Strategy

A systematic search was performed in six databases (Web of Science, Scopus, SportDiscus, Pubmed, Eric, and PsycInfo) during the month of December 2021. Three blocks were envisaged to elaborate upon the search phrase: (1) adapted sport OR disability sport OR Paralympic sports OR Paralympic games; (2) Paralympic OR Paralympic athletes OR disabled athletes OR disability OR impairment; and (3) social context OR environmental context OR training process OR training OR psychological aspects OR psychological skills OR technical aspects OR tactical aspects OR technical-tactical aspects OR physical fitness OR physical condition. The blocks were combined with the Bolean operator AND. For the development of the search phrase, the thematic blocks were established beforehand. Subsequently, after a preliminary search in the different databases and in different articles on the topic under study, the terms of the phrase were added in each thematic block together with their corresponding synonyms. The search phrase was entered in English only. It should be noted that the search was not limited to a specific date.

### 2.3. Study Selection and Data Extraction Process

Once the database search was completed, the titles and abstracts of the documents obtained were analysed in order to select those which were directly related to the subject matter of the study and to eliminate those which did not meet the inclusion criteria. The population, intervention, control group, and outcome (PICO) components were considered. After this screening process, the articles were selected for further data collection. A total of 19 articles were then selected: 6 from Web of Science, 8 from SportDiscus, and 4 from PsycInfo. No articles meeting the inclusion criteria were found in the Scopus, Eric, or Pubmed databases. In addition, one article was selected by reviewing the references of the articles included in the review.

### 2.4. Quality Assessment

Quality assessment of the selected manuscripts was undertaken using the Standard Quality Assessment Criteria for quantitative and qualitative studies [23]. This tool is based on a systematic scoring system for evaluating the quality of studies and serves to ensure a minimum of quality in a systematic review [23]. For quantitative studies, 14 items were scored (“yes” = 2, “partial” = 1, and “no” = 0). Items not applicable to a particular study were marked “n/a” and were excluded from the calculation of the summation score. The scores of the qualitative investigations were estimated in a similar way, considering the scores obtained in 10 items. Inter-observer agreement was calculated using the intra-class correlation coefficient, resulting in a near perfect coefficient of 885 (*p* < 0.05) [24]. Two researchers assessed the quality of the articles independently. In case of discrepancies, these were resolved by the third researcher.

## 3. Results

### 3.1. Selection of Studies

A total of 2392 results were found in the initial search, and 346 duplicate articles were removed using Excel version 2020, leaving 2046. Subsequently, 1936 manuscripts were excluded after screening for the title and abstract. Ninety-two were also removed as they were systematic or literature reviews. In addition, one article was selected by reviewing the references of the included articles. Finally, 19 articles were included in this systematic review, as they met all inclusion criteria (see Figure 1).

### 3.2. Quality of the Studies

The scores for research quality were expressed as percentages and are shown in Table 1 and Table 2. The overall scores assigned by Rater 1 for qualitative and quantitative research ranged between 70% and 90%, and between 75% and 100%, respectively, while Rater 2’s scores ranged between 70% and 100% for qualitative studies and between 75% and 100% for quantitative studies. For the inclusion of articles, a cut-off point of 70% was established for all studies.

### 3.3. Characteristics of the Studies

The characteristics of the participants, sports, and factors of the studies included in the systematic review are shown in Table 3 and Table 4 below. Furthermore, the most salient aspects of the training process of Paralympic athletes are shown in Table 5.

## 4. Discussion

The aim of this systematic review was to identify the factors influencing the training process of para-athletes, as well as the barriers they encounter. For this purpose, the review carried out included only research of optimal quality. In this sense, it should be noted that all the articles included received a score equal to or greater than the cut-off point (≥70%) at the beginning of the evaluation process, which may be due to the requirement of the inclusion and exclusion criteria defined previously. In this way, the tool used to assess the quality of the documents can help to identify differences between the studies and to synthesise and interpret the main results of the studies [23].

The main results showed that the factors influencing the training process of Paralympic athletes referred to the sport context, social context, psychological aspects, technical-tactical contents, and physical condition. In addition, numerous barriers were found in the training and development process of these athletes.

The training and performance of Paralympic athletes, in relation to the sport context, is influenced by several aspects [16], among which the figure of the coach stands out both for the support they offer to the athletes and the relationship they establish with them, as well as the qualifications they possess [17,38]. Thus, the coach exerts a decisive influence on athletes with a disability [74] and is a fundamental and essential part of their sports career [75]. In this regard, Banack et al. [76] reported that athletes who perceive the support of the coach are more motivated, which generates greater persistence, enjoyment, and effort. Vella et al. [77] affirmed that a caring relationship between the athlete and the coach generates a positive atmosphere, and this positively affects performance, as athletes consider the figure of the coach as a reference [78].

Sport facilities are an essential element in the sport context, but they can become an obstacle for athletes with disabilities [16,31]. Ellis et al. [79] and Shirazipour et al. [80] considered that many facilities do not offer accessible services, in many cases making it difficult for athletes to move from one location to another. Díaz et al. [81] also reflected on the role of sports equipment for people with disabilities, and they concluded that sports equipment for these people is very expensive, and it is generally customised according to the needs of each person, so they can practically only be used by one athlete.

In terms of the social context, the relationship between Paralympic athletes and their friendships is unquestionable [32]. Sport is considered a normalising agent because it can become the ideal vehicle for people with a disability to establish and strengthen social relationships [82]. In this sense, Imms [83] stated that for these athletes, support from friends, teammates, or national teammates is a key issue to continue participating in sports. Family also plays a crucial role in the participation of athletes with a disability [29], which is consistent with the findings of other studies [84,85]. Robinson et al. [86] considered that participation in sports for Paralympic athletes requires continuous commitment from both the athlete and the family.

Regarding the psychological factor, the practice of physical activity and sport is associated with mental health benefits [87,88]. Tasiemski et al. [89] concluded that people with a disability who are physically active have higher levels of life satisfaction and, as a consequence, lower levels of stress. Along these lines, Puce et al. [90] stated that there is a series of psychological mechanisms triggered by the practice of sport, which can contribute to a reduction in both stress and anxiety. On the other hand, it should be noted that psychological aspects are determinant in achieving peak performance [25], although they are sometimes not considered as important as they deserve to be [91]. The results also highlighted the relevance of self-esteem [30,31,34,39]. Thus, Marsh et al. [92] stated that positive self-esteem facilitates the desired achievement, which is to win a medal at the Paralympic Games. Moreover, the level of self-esteem is similar between Paralympic athletes and athletes from the general population [93]. Pensgaard et al. [94] pointed out the high levels of resilience of Paralympic athletes, who are considered resilient people because they have the ability to overcome mistakes and believe that they can achieve success [95]. Psychological studies on Paralympic sport have mainly focused on psychosocial effects, but it also has other benefits related to the reduction of depression [96], which is underlined in the study by Martin et al. [33]. This is relevant as people with disabilities are often prone to suffering from this type of mental disorder [97]. Furthermore, the results highlight that sport practice reduces anxiety in athletes with disabilities [30,32,39] as well as stress [30,32,35,39]. Nevertheless, sport can also be the source of these psychological problems, as elite athletes face numerous stressors such as anxiety, pain, or fear of not achieving success [98]. In this regard, Wilson et al. [99] conducted a study in which they observed that gender has an influence, as female athletes were more likely to experience anxiety during preparation for the Paralympic Games than male athletes [99]. In terms of motivation, Szájer et al. [39] reported higher scores in Paralympic athletes than in non-disabled athletes. Perhaps, as De Guast et al. [100] considered, a Paralympic athlete uses his or her disability as a source of motivation and self-improvement. In addition, Shapiro and Martin [101] stated that motivation in people with disabilities is strongly associated with the competitive world. Thus, Paralympic athletes who tend to be highly motivated achieve success at a higher rate than those who are not [76].

The technical-tactical factor was underlined by Van Biesen et al. [40] in relation to the training process of Paralympic athletes. In this respect, Marszalek et al. [102] stated that it is necessary to look for the relationship between performance, enjoyment, health, and technical-tactical factors. In elite sports, technical-tactical aspects are of great importance, as the aim is for the athlete to achieve optimal performance, which entails improving these factors during training, with the aim of achieving success during competition at the Paralympic Games [103]. Again, the role of the coach is essential [104].

Another important factor in the training of Paralympic athletes is physical condition [27,35,36]. For Jackson [105], the parameters determining physical condition are strength, speed, flexibility, balance, agility and cardiorespiratory endurance, and body composition, which have a decisive influence not only on the physical condition of athletes with a disability but also on sport performance [106]. However, Paralympic athletes tend to be very susceptible to early fatigue [10] and have higher metabolic costs [107], so addressing these aspects in the training and performance of these athletes is crucial.

Gender and disability barriers are also a major factor affecting the training and performance of Paralympic athletes [16,17,28,31]. Women often do not have access to sports as easily as men, especially at the competitive level [108]. Furthermore, women’s participation in sports has been criticised and rejected [109]. For this reason, women in sports have had no choice but to try to resist these mechanisms of marginalisation [110]. Nonetheless, women with disabilities are increasingly present in both recreational physical activity and elite sports [111]. On the other hand, disability barriers directly influence the usual levels of physical activity [16] and, as a consequence, the participation levels of people with disabilities [112]. The literature has shown stereotypes about disabilities and described how people with disabilities are discredited, devalued, and even humiliated for not conforming to, among other things, normal physical activity and sport [113]. For these reasons, people with disabilities feel excluded or alienated from sporting activities and may also feel frustrated by being treated differently [114,115,116], which may explain their low level of participation [117]. In this regard, Cid [118] stated that throughout history, persons with disabilities have been subjected to rejection and discrimination, and they are usually relegated to the background in society. At present, however, progress is evident, and we can see how these people are integrating into the world of work and, little by little, are overcoming some of the many barriers that affect them. She concludes by saying that it is essential to continue this work until all persons with disabilities fully enjoy their rights to integrity, education, work, non-discrimination, and full integration into society.

Another aspect to highlight is the economic barriers [16,25,28,29], since the lack of sponsorship in Paralympic sport means that athletes are often responsible for paying most of the expenses involved in elite sports. The media can also be a barrier in Paralympic sport [28,29]. In this vein, Marín [119] pointed out that it is now clear that sporting activity, especially elite sports, has achieved great interest from the economic and commercial point of view, as well as from the media’s follow-up. This author continues to comment that the professionalisation of sport has led to the commercialisation of the athlete’s image through activities such as marketing, merchandising, image rights, or media presence. Businesses and the media tend to focus on sports that have the greatest impact on television. According to Rimmer [120], both the media and marketing are barriers in the social context, which considerably limit the sporting participation of people with disabilities due to the lack of visibility of these athletes.

The findings of the review indicate that some disabilities have been studied more than others. In this regard, visual and physical disabilities were highlighted. In addition, there were differences in the factors and barriers studied according to the type of disability. In this way, and with regard to visual impairment, the main considerations referred to the fact that the practice of sport had a great impact on the self-confidence and self-esteem of athletes, as well as the need to work on the control of pressure and stress. In addition, the main barriers faced by the visually impaired were environmental and financial as well as poor media coverage, and their main demands were for greater social recognition, increased economic benefits, as well as better health care [16,25,26,28,30,31,34,35,37,38]. As for para-athletes with amputations, it should be noted that the main barriers faced by them were environmental ones, and personal factors were their main facilitators [25,31,36]. Studies in para-athletes with cerebral palsy revealed the need to work on some psychological aspects such as self-confidence, motivation, and anxiety control. In addition, the main facilitators were the people around them, the desire to win a competition, and the socialisation provided by the practice of sport [25,32,39]. In people with hearing impairment, the practice of sport contributed positively to self-confidence, self-esteem, and control of anxiety, while the main facilitators were related to the people around them, the desire to win a competition, and socialisation [30,32]. Regarding para-athletes with some physical disability, the practice of sport was also positive. However, it is recommended to continue working on controlling the pressure exerted by sports, self-confidence, motivation, and anxiety. This question is important because psychological care in para-athletes is not usually common [38]. On the other hand, the main barriers faced by persons with physical disabilities were environmental and financial, along with poor media coverage and low recognition as professional athletes. The main facilitators were family, socialisation, enjoyment of sports, and improved self-perception. However, these athletes demanded greater social and economic benefits [16,28,29,30,33,34,35,37,38,39]. As for people with spinal cord or neurological injuries, they considered that the main barriers they encountered were the environmental ones, while the facilitators had to do with personal factors [31]. With regard to people with intellectual disabilities, the studies revealed the need to work on the control of the pressure exerted by the practice of sports on people and sports technique [35,40]. Finally, the study by Sánchez-Pay and Sanz-Rivas [36] showed the need to work on physical condition in athletes with osteogenesis. In summary, the above considerations should be taken into account in the training process of para-athletes in order to improve or optimise the training process.

The systematic review conducted has some limitations. First is the existence of few studies on the subject analysed. In addition, the search was limited to three languages—Spanish, English, and Portuguese—and it was limited to full-text articles. In the review carried out, the factors that influence the training of Paralympic athletes and the barriers they encounter in general were analysed without specifically taking into account the sport practiced or the disabilities that the athletes had. Therefore, future research could focus on analysing the differentiated and comparative training process between female and male Paralympic athletes as well as analyse the factors that influence and limit this process according to the sports or disabilities presented by the athletes.

With the aim of answering the questions raised at the beginning of the manuscript, it must be said that the factors influencing the training process of Paralympic athletes refer to the sporting, social, psychological, technical-tactical, and physical condition spheres. Furthermore, regarding the main barriers faced by Paralympic athletes, the need for more financial and technical support, invisibility in the media, disability-related barriers, and dependence on other people stand out.

## 5. Conclusions

In the sports training process of para-athletes, contextual (sporting and social), psychological, technical-tactical, and physical condition factors can be distinguished. Among the factors related to the sports context, the importance of the figure of the coach, his or her training, and the relationship maintained with the athletes stand out. Social factors also play an important role, such as the relationships athletes have with each other and the support of their families. In terms of psychological factors, it is worth mentioning the importance of self-esteem and motivation, as well as stress reduction. Factors related to physical condition, technique, and tactics are also relevant. On the other hand, the training process of Paralympic athletes is influenced to a large extent by numerous barriers, among which, on the one hand, the economic aspects can be highlighted, as the material, transport, and facilities they need for sports are very expensive and, on the other hand, the visibility in the media, as Paralympic sports always appear in the background, which means that the successes are not so socially recognised and that the support received by these athletes is minimal.

In summary, the results obtained in this systematic review can be of great help to coaches and competent institutions in the field (Paralympic committees, federations, etc.) in order to improve the training process and the performance of para-athletes, as well as eliminate the barriers that this group encounters and promote policies which facilitate access to sport for people with disabilities.

## Figures and Tables

**Figure 1 ijerph-19-07242-f001:**
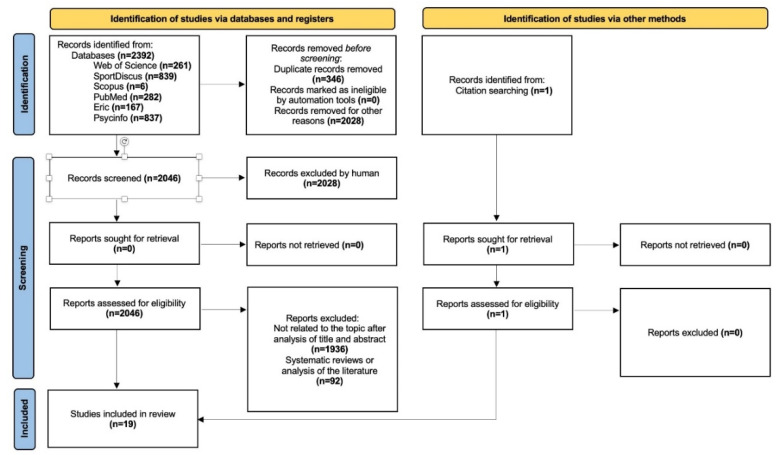
Flowchart of the systematic review process according to the PRISMA protocol declarations.

**Table 1 ijerph-19-07242-t001:** Quality assessment of qualitative studies.

Studies	Rater 1 (%)	Rater 2 (%)
Alexander et al. [17]	90	90
Arnold et al. [25]	80	80
Eddy and Mellalieu [26]	85	100
Page et al. [27]	70	70
Rodrigues et al. [28]	80	70
Vieira et al. [29]	80	70

**Table 2 ijerph-19-07242-t002:** Quality assessment of quantitative studies.

Studies	Rater 1 (%)	Rater 2 (%)
Baĉanac et al. [30]	75	75
Jaarsma et al. [31]	90	100
Kokun and Shamych [32]	.86	88
Martin et al. [33]	95	100
Marín-Urquiza et al. [34]	100	100
Pereira et al. [35]	86	85
Sánchez-Pay and Sanz-Rivas [36]	90	85
Sobiecka et al. [37]	90	100
Sobiecka et al. [38]	95	100
Sobiecka et al. [16]	95	100
Szájer et al. [39]	95	100
Van Biesen et al. [40]	90	90
Wood et al. [41]	86	85

**Table 3 ijerph-19-07242-t003:** Participant characteristics, sports, and factors of the studies included in the systematic review.

Studies	Country	Subjects	Age (M ± SD)	Sports	Disability	Factors
Alexander et al. [17]	Canada	8 women	NR	NR	NR	Sport context Barriers
Arnold et al. [25]	NR	18 men	25.44 ± 6.41	Swimming, paratriathlon, athletics, cycling, golf, basketball, and cricket	Visual Amputation Cerebral palsy	Social context Sport context Barriers
Baĉanac et al. [30]	NR	24 12 with disability (10 men) (2 women) 12 without disability (10 men) (2 women)	With disability (28.58 ± 5.71) Without disability (26.21 ± 6.36)	With disability (athletics, archery, table tennis, chess, and cycling) Without disability (athletics, table tennis, archery, and cycling)	Visual Hearing Physical	Psychological
Eddy and Mellalieu [26]	Great Britain	6 (women)	21.8 ± 8.0	Goalball	Visual	Psychological
Jaarsma et al. [31]	The Netherlands	76 (20 men) (46 women)	30.5 ± 9.7	Alpine skiing, athletics, archery, equestrian, rowing, tennis, table tennis, 7-a-side football, cycling, volleyball, and swimming	Visual Amputation Spinal cord injury Cerebral palsy Neurological	Social context Sport context Barriers and facilitators
Kokun and Shamych [32]	Ukraine	36 men	NR	NR	Hearing impairment Cerebral palsy	Psychological
Martin et al. [33]	United States	25 women	25.80 ± 5.24	Wheelchair basketball	Physical Spinal cord injury Spina Bifida	Psychological
Marín-Urquiza et al. [34]	Belgium, Ireland, United Kingdom, United States, Australia, Canada, Portugal, Spain, and Argentina	84 (69 men) (15 women)	NR	Alpine skiing, athletics, pétanque, cycling, indoor football, goalball, judo, pentathlon, rowing, sailing, archery, sitting volleyball, swimming, table tennis, triathlon, wheelchair basketball, wheelchair rugby, and powerlifting	Visual Physical	Psychological
Page et al. [27]	United States	6 (2 men) (6 women)	NR	wheelchair fencing, powerlifting, swimming, athletics, and wheelchair basketball	NR	Psychological Social context Physical condition
Pereira et al. [35]	Brazil	11 men	22.73 ± 5.00	Swimming	Visual Physical Intellectual	Psychological
Rodrigues et al. [28]	Portugal	9 (8 men) (1 woman)	NR	Swimming, boccia	Visual Physical	Social context Barriers
Sánchez-Pay and Sanz-Rivas [36]	NR	9 men	38.35 ± 11.28	Wheelchair tennis	Amputation Spinal cord injury Osteogenesis	Physical condition
Sobiecka et al. [37]	Poland	89 (58 men) (31 women)	Men 32 ± 9.7 Women 32 ± 9.32	Horse riding, cycling, athletics, athletics, archery, swimming, fencing, tennis, table tennis, and rowing	Physical Visual	Psychological Social context Sport
Sobiecka et al. [38]	Poland	91 (61 men) (30 women)	NR	NR	Physical Visual	Psychological Social context Sport context Barriers
Sobiecka et al. [16]	Poland	470 Group 1 (324) (254 men) (70 women) Group 2 (146) (118 men) (28 women)	Group 1 Men (32 ± 11.0) Women 28 ± 8.20 Group 2 Men (32 ± 12.10) Women (33 ± 12.30)	NR	Physical Visual Physical	Psychological Social context Sport context Barriers
Szájer et al. [39]	Hungary	18 with disability (9 men) (9 women) 35 without disability (23 men) (12 women)	With disability (26.33 ± 10.81) Without disability (23.26 ± 3.85)	Swimming	Physical Cerebral palsy	Psychological
Van Biesen et al. [40]	Czech Republic	71 with disability (41 men) (30 women) 17 without disability (12 men) (5 women)	With disability Men 27.00 ± 8.00 Women 28.00 ± 8.00 Without disability Men 24.00 ± 12.00 Women 20.00 ± 9.00	Table tennis	Intellectual	Technical-tactical
Vieria et al. [29]	Brazil	9 (6 men) (3 women)	NR	Boccia, athletics, volleyball, goalball, swimming, and paracanoeing	Physical	Barriers and facilitators
Wood et al. [41]	NR	8 (5 men) (6 women)	40.12 ± 12.99	NR	NR	Psychological Sport context Barriers

**Table 4 ijerph-19-07242-t004:** Methodology, objectives, and main findings of the studies included in the systematic review.

Studies	Methodology	Instrument	Objectives	Main Findings
Alexander et al. [17]	Qualitative	Semi-structured and one-to-one interviews [42,43]	Explore the perceptions and experiences of athletes competing in an individual or coercive sport	All athletes made significant sporting achievements and recognised the importance and value of their coaches in helping them reach high standards of success. In addition, participants highlighted that male coaches inappropriately address their disability and gender issues and how this influenced their psychological well-being.
Arnold et al. [25]	Qualitative	Semi-structured and one-to-one interviews [44,45]	Explore the different organisational stressors faced by athletes with a disability	A total of 316 organisational stressors were identified, which were summarised into 31 concepts and 4 pre-conceptualised exploratory schemes: leadership and staff issues, cultural and team issues, logistical and environmental issues, and performance and personal issues
Baĉanac et al. [30]	Quantitative	Self-Esteem Scale (RSES) [46] Sport Confidence Scale (SC) [47] Athletic Coping Skills Inventory-28 [48] Sport Competitive Anxiety Test (SCATr) [49]	Prove that participation in sport contributes to the psychological improvement of people with disabilities, as well as help them develop positive attitudes towards themselves and their general and sporting competence and become more able to cope with stress	The psychological profile of athletes with disabilities is very similar to the profile of athletes without them, which shows that sports positively contribute to their physical strength, making them equally prepared for the best results in sports as athletes without disabilities. The practice of sport has a positive impact not only on self-confidence in sports but also on the overall self-esteem of athletes with disabilities. Their anxiety about competing is optimised, and their psychological abilities to cope with stress are improved, making them no different from their non-disabled peers.
Eddy and Mellalieu [26]	Qualitative	Structured interview developed by the study’s authors	Research the imagery of experiences of visually impaired Paralympic athletes	Participants reported using imagery for cognitive and motivational purposes in both training and competition
Jaarsma et al. [31]	Quantitative	Questionnaire partially based on the questionnaire validated by the Mulier Instituut [50]	Understand the barriers and facilitators of sport for Paralympic athletes	This study indicated that barriers in sport were mainly environmental, while facilitators were generally personal factors. Attitude and subjective rules were considered to be the most important components of the intention to participate in sport.
Kokun and Shamych [32]	Quantitative	Questionnaire developed by the study authors Self-efficacy scale [51] Self-esteem scales [52]	Determine the characteristics and common factors of Paralympic athletes’ self-fulfilment	The results obtained suggest that the most significant incentive for Paralympic athletes to engage in sports is the moral satisfaction of winning a competition. Furthermore, there are other important incentives such as opportunities for personal self-fulfilment, socialising with friends, the ability to be a full member of society, gaining experience, meeting new people who could help later in life, opportunities or prospects for travel abroad, and the pleasure of training.
Martin et al. [33]	Quantitative	Profile of Mood State Questionnaire (POMS) [53] Sixteen-Factor Personality Questionnaire (16PF) [54], but the fifth edition was used [55]	Determine whether personality and mood differences existed between elite Paralympic athletes and elite athletes who did not qualify for the team	The results revealed that athletes who were part of the Paralympic team scored higher on toughness and lower on anxiety. In terms of mood, Paralympic athletes scored higher in vigour and lower in depressed mood.
Marín-Urquiza et al. [34]	Quantitative	Athletic Identity Measurement Scale (AIMS) [56] Rosenberg Self-Esteem Scale (RSES) [46] Perceived Stress Scale (PSS) [57]	Provide more information on Paralympic sport transition and its impact on athletes	No differences in self-esteem were found between the withdrawn group and the active group. However, within the withdrawn group, athletes who withdrew involuntarily had significantly lower self-esteem scores than those who withdrew voluntarily.
Page et al. [27]	Qualitative	Qualitative Sports Orientation Questionnaire [58]	Researching the reasons why six elite athletes with disabilities play sports	The results revealed a desire to abandon or at least temporarily escape from the social and physical conditions associated with disability. Specifically, participants discussed the desire to gain competence for oneself and with others through participation in sport and the desire to stay active to fight the combined effects of being inactive or being disabled. Additionally, there was the desire to participate in a social activity with other people with disabilities.
Pereira et al. [35]	Quantitative	Sport Motivation Scale [59] Recovery-Stress Questionnaire for Athletes (RESTQ76-Sport) [60]	Research the associations between the hormonal profile and the psycho-biological aspects of Paralympic athletes during a competition season	No significant differences were found between motivation levels, but significant differences were found in terms of pressure.
Rodrigues et al. [28]	Qualitative	Semi-structured individual interviews and methods in [61]	Identify the characteristics and trends of the media approach to the Paralympic Movement	The results showed that, in many cases, Paralympic athletes are unhappy with the limited media coverage, the stigma, and the supremacy of football, and there is an evident desire to occupy a more important space in the sporting arena, which confers greater legitimacy and the possibility of athletic-competitive development, as well as economic and social gains
Sánchez-Pay and Sanz-Rivas [36]	Quantitative	Field tests widely used in the assessment of tennis players and wheelchair players [62,63,64,65]	Measure the fitness levels of wheelchair tennis players and make comparisons according to ranking and type of injury	The results showed that higher-level players have better fitness levels in all tests, finding significant differences in almost all tests. Players with a higher functional limitation showed lower values, although no statistically significant differences were found.
Sobiecka et al. [37]	Quantitative	Interview for male and female athletes training for the Paralympic Games [66]	Evaluate the preparation process of Polish athletes with disabilities for the Beijing 2008 Paralympic Summer Games	The results revealed that the athletes were fully satisfied with the camps (food, accommodation, and sports equipment) and social relations (atmosphere of cooperation between athletes and cooperation between athletes and coaches of national teams of different disciplines). However, they require more attention in health (medical care and individual orthopaedic equipment) and contact with the media.
Sobiecka et al. [38]	Quantitative	Adjusted questionnaire for disabled sports [66]	Present the conditions during the preparations of Polish athletes for the Paralympic Summer Games (2004–2012)	The analysis showed that relationships between athletes were good at all times, but accommodations and food were poor. Cooperation with doctors, physiotherapists, and masseurs was satisfactory. Consultations with the dietician were sporadic and evaluated as poor. Consultations with sport psychologists were rare, but satisfactory.
Sobiecka et al. [16]	Quantitative	Form for male and female athletes of the national team [16]	Identify the limitations observed in Polish Paralympic sports in terms of the environment in which athletes train on a daily basis	Particularly outstanding difficulties were related to organisational and financial issues. At the same time, the environment was shown to be a differentiating factor.
Szájer et al. [39]	Quantitative	Adaptation of the Competitive State Anxiety Inventory-2 (CSAI-2) [67] from the original [68] Adaptation of the Athletic Coping Skills Inventory-28 (ACSI-28) [69] from the original [48]	Reveal possible differences in several psychological profiles between Paralympic swimmers and healthy swimmers. In addition, another objective was to explore possible gender differences and differences between successful and less successful swimmers.	Healthy swimmers scored lower on somatic anxiety and higher on self-confidence, absence of worry, and self-confidence or motivation for achievement than Paralympic swimmers. When the tests were repeated separately for men and women, the results remained unchanged for women, while healthy male athletes only scored significantly higher than male Paralympic swimmers in self-confidence and absence of worry. In addition, medal-winning athletes in the overall sample exhibited less cognitive and somatic anxiety. Para-swimmers with different levels of disability did not differ from each other in any of the measures. The findings show that para-swimmers experience significant psychological disadvantages.
Van Biesen et al. [40]	Quantitative	Ten different series of 10 identical strokes	Determine the technical competence of table tennis players with and without intellectual disabilities	Statistical analysis suggested that there were no gender differences in terms of proficiency. Table tennis players without a disability scored significantly better than those without a disability.
Vieria et al. [29]	Qualitative	Semi-structured interview [70]	Explore facilitators and barriers to the practice of sports by high-performance Brazilian athletes with disabilities actively involved in national or international competitions	The main facilitators mentioned were family support, socialisation, economic benefits of sport, incentives from rehabilitation centres and health professionals, the possibility of visiting new places, enjoyment of sport and competition, better perception of their abilities, development of autonomy, and access to places offering free adapted sports. The most frequently cited barriers were physical wear and tear, lack of appreciation and recognition of athletes as sports professionals, lack of sponsorship, accessibility difficulties, and the lack of media coverage.
Wood et al. [41]	Quantitative	Shortened General Attitudes and Beliefs Scale (SGABS) [71] State Trait Personality Inventory (STPI) [72] Achievement goals questionnaire (AGQ) [73]	Examine the immediate effects of rational emotive behavioural therapy on psychological, physiological, and performance outcomes with elite Paralympic athletes	Visual and statistical analyses of the data indicated that reductions in irrational beliefs were combined with reductions in systolic blood pressure, indicative of an adaptive physiological response, improved athletic performance during competition simulations, and reductions in goal avoidance. In addition, social validation data showed increased self-awareness, emotional control, and greater concentration during competition as a result of the rational emotive behavioural therapy intervention.

**Table 5 ijerph-19-07242-t005:** Outstanding aspects in the training process of para-athletes.

Sport Context	Social Context	Psychological Aspects	Technical-Tactical Aspects	Physical Condition
Importance of the figure of the coach [17] Relationship with coaches [16,17,32,37] Necessary training of coaches [16,31] Relevance of sport-specific facilities [25,37,41] Importance of having a medical team [35] Importance of competing and winning [31] Barriers: Need to adapt sports facilities and increase the number of sports facilities [16,31] Suitability of a medical team [16,38] Need for more financial support [16,25,28]	Significance of the relationship between athletes [25,27,31] Importance of the family [29,31] Barriers: By disability [16,31] By gender [17,28] Dependence on others [31] Lack of visibility in the media [28,29] Lack of sponsors [16] and lack of facilities, supervision, and transport [31]	Relevance of self-esteem [30,31,34,39] Importance of health [31] Reduction of stress [30,33,35,39] and depression [33] Significance of motivation [26,30,31,35] Barriers: Excessive stress [41]	Relevance of technical-tactical aspects [40]	Importance of physical condition [27,35,36]

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
