# Peer review of "The Sport Training Process of Para-Athletes: A Systematic Review"

_ijerph, 2022, doi:10.3390/ijerph19127242_

Round 1
Reviewer 1 Report
Dear authors,
I think the manuscript is clear and well-written. The main purpose is very interesting. The methods are clear and future researchers could reproduce your study. The results obtained in this systematic review can be of great help not only to experts in the field but also to future research projects. I think it will be important in the future to better understand the mechanisms related to the effects induced by specific training without neglecting the social and political importance
Author Response
Dear reviewer,
we appreciate your comments and the evaluation of the work, which encourages us to continue working along the same lines.
Thank you very much.
Reviewer 2 Report
Dear Authors,
After a careful reading of the manuscript titled "The sport training process of Paralympic athletes. A systematic review" I consider that the topic is relevant for the development of better and more fullfil conditions for practicing sports for the population with disabilities in general and in particular for Paralympic athletes. The aim of this systematic review was to determine the factors influencing the training process of Paralympic athletes, as well as the barriers they encounter. For this reason, the present study allows to increase knowledge about the factors influencing the training process of Paralympic athletes, as well as to identify the barriers encountered by these athletes in this process.
I have some comments and suggestions for improving the manuscript' quality which I mention below:
# please complete the affiliation details,e.g., university name, address, city and country.
Introduction section
# in the introduction section I suggest that reference be made to whether there are differences in the type of influences and constraints related to each sport on the training process of Paralympic athletes. The question that arises is whether the type of influences and constraints are the same in any Paralympic modality, since the contexts are different. An example is the difference in sport facilities between modalities and which are an essential element in the sport context.
Section 2. Materials and Methods
# Please indicate the revision registration number protocoled at the International Prospective Register of Systematic Reviews (PROSPERO)
# Please indicate whether the authors performed the data search by using the U.S. National Library of Medicine’s Medical Subject Headings terms and English language terms related to the sport training process and the Paralympic athletes population.
#please indicate how many different searches in each of the mentioned databases it was possible to perform through word combinations.
# the eligibility criteria of item "b) no theses, book chapters or conference proceedings" is an exclusion criterion, please withdraw or amend
# lines 139 and 140 - section "Quality of the studies" the authors state that "The scores for research quality were expressed as percentages and are shown in Tables 1 and 2." I ask the authors to confirm that the values ​​are expressed as a percentage.
#The inter-rater agreement for paper inclusion/exclusion uses the variety of cut-points for the overall scores in quantitative studies and the quality scores is also used to define the minimum threshold for inclusion of studies in the systematic review. Please indicate which cut-points are selected for articles inclusion.
# please avoid putting tables in a row without introductory text, I refer to tables 3, 4 and 5.
# in the discussion it is not clear whether the factors influencing the training process of Paralympic athletes are the same regardless of the modality that is played. Please address this issue.
BW
Author Response
COVER LETTER
Manuscript ID: ijerph-1747418. Type of manuscript: Article. Title: The sport training process of Paralympic athletes. A systematic review.
Reviewer 2’s comments and suggestions for authors |
Details of the revisions and responses |
Please complete the affiliation details,e.g., university name, address, city and country. |
The affiliation details of the authors' affiliation have been added. |
In the introduction section I suggest that reference be made to whether there are differences in the type of influences and constraints related to each sport on the training process of Paralympic athletes. The question that arises is whether the type of influences and constraints are the same in any Paralympic modality, since the contexts are different. An example is the difference in sport facilities between modalities and which are an essential element in the sport context. |
The following clarifying paragraph has been included in the introduction: In this regard, it should be borne in mind that both influences and constraints may de-pend on the type of sport in question, which could be the focus of future research. |
Please indicate the revision registration number protocoled at the International Prospective Register of Systematic Reviews (PROSPERO) |
PROSPERO does not accept records of protocols that do not include health-related outcomes. We have tried to register the revision on the Inplasy platform, but it is not open access. We will take this into account from the outset for future work. |
Please indicate whether the authors performed the data search by using the U.S. National Library of Medicine’s Medical Subject Headings terms and English language terms related to the sport training process and the Paralympic athletes population. |
The search for data was based on the terms found in numerous studies related to the training process of Paralympic athletes and the barriers encountered by them. |
Please indicate how many different searches in each of the mentioned databases it was possible to perform through word combinations. |
In the search phrase, 3 thematic blocks were established beforehand. Subsequently, different articles and investigations were searched with the intention of completing the sentence with all the appropriate terms and their synonyms. Finally, the same search phrase was used in all databases. |
The eligibility criteria of item "b) no theses, book chapters or conference proceedings" is an exclusion criterion, please withdraw or amend |
Modified taking reviewer's suggestion into account. The inclusion criteria used for the selection of manuscripts were as follows: a) full-text articles; b) subjects had to be Paralympic athletes; c) manuscripts had to be written in English, Spanish or Portuguese. On the other hand, the exclusion criteria were a) studies not related to the training process of Paralympic athletes; b) no systematic reviews or literature reviews; c) no theses, book chapters or conference proceedings. |
lines 139 and 140 - section "Quality of the studies" the authors state that "The scores for research quality were expressed as percentages and are shown in Tables 1 and 2." I ask the authors to confirm that the values are expressed as a percentage. |
Modified taking reviewer's suggestion into account. |
The inter-rater agreement for paper inclusion/exclusion uses the variety of cut-points for the overall scores in quantitative studies and the quality scores is also used to define the minimum threshold for inclusion of studies in the systematic review. Please indicate which cut-points are selected for articles inclusion. |
The requested information has been included: For the inclusion of articles, a cut-off point of 70% was established for all studies. |
Please avoid putting tables in a row without introductory text, I refer to tables 3, 4 and 5. |
The following text appears just before the Tables.
The characteristics of the participants, sports and factors of the studies included in the systematic review are shown in Tables 3 and 4 below. Furthermore, the most salient aspects of the training process of Paralympic athletes are shown in Table 5.
We have tried to put text right before each Table, but it gives us formatting problems. We’ll talk to the publisher to correct it. |
In the discussion it is not clear whether the factors influencing the training process of Paralympic athletes are the same regardless of the modality that is played. Please address this issue. |
In the review carried out, the factors that influence the training of Paralympic athletes and the barriers they encounter in general have been analyzed without specifically taking into account the sport practiced or the disability that the athletes had. This question has been contemplated both in the limitations and in the future investigations of the study carried out. In the review carried out, the factors that influence the training of Paralympic athletes and the barriers they encounter in general have been analyzed without specifically taking into account the sport practiced or the disability that the athletes had. So, future research could focus on analysing the differentiated and comparative training process between female and male Paralympic athletes, as well as to analyse the factors that in-fluence and limit this process according to the sport and/or disability presented by the athletes. |
THANK YOU for your comments and suggestions.

Reviewer 3 Report
This systematic review aimed to summarize significant parameters and barriers that may affect the training process in disabled athletes. For this purpose, the authors have searched 6 databases using the PRISMA protocol, which is a strength of the paper. Here some questions and suggestions to improve the paper:
title: paralympic athletes may give the impression that only athletes competing at the olympic games were subject of interest, which is not the case. Suggestion to name the target population differently.
There is wide range of possible disabilities, which is also reported in the results section. There is a reasonable assumption that the parameters/barriers may differ depending on the kind of disability. Therefore I suggest to get a closer look on that, and integrate this in the discussion section.
Quality assessment was performed, but this important aspect of a systematic review is not discussed. Suggestion to integrate the quality of the studies and forthcoming findings in the discussion section.
line 40-41: 2x with a disability in same sentence
line 50: correct way of referring? (11)
line 87: no systematic review as exclusion, (leave 'no' out) otherwise you say that SR's were included.
line 93-94: this info was already provided supra. delete here.
line 96: not clear if these 3 'blocks' should reflect the 'PECO' search, and if these were combined (using 'AND' operator?)?
line 117-118: explain shortly how this tool works.
line 121: 'was' should be 'were'
line 124: duplicates were removed. Is this mentionned in the methods section? Which software was used?
line 171: please find a different way than to use 'sporting life', maybe 'sports career'?
line 252: can you better clarify the link between economics and media attention?
line 256: That was 20 years ago. A lot of things may have changed since then. Therefore it would be interesting to see if there is some evolution in parameters, which should be possible due to no year of publication restriction in your search. Anyway, the evolution over time considering this subject should be discussed.
Author Response
COVER LETTER
Manuscript ID: ijerph-1747418. Type of manuscript: Article. Title: The sport training process of Paralympic athletes. A systematic review.
Reviewer 3’s comments and suggestions for authors |
Details of the revisions and responses |
Title: paralympic athletes may give the impression that only athletes competing at the olympic games were subject of interest, which is not the case. Suggestion to name the target population differently. |
We have no problem changing the title of the article, but the study carried out has focused on Palalympic athletes. In fact, an inclusion criterion was that the studies focus on Paralympic athletes. |
There is wide range of possible disabilities, which is also reported in the results section. There is a reasonable assumption that the parameters/barriers may differ depending on the kind of disability. Therefore I suggest to get a closer look on that, and integrate this in the discussion section. |
In the review carried out, the factors that influence the training of Paralympic athletes and the barriers they encounter in general have been analyzed without specifically taking into account the sport practiced or the disability that the athletes had. In addition, this question has been contemplated both in the limitations and in the future investigations of the study carried out. In the review carried out, the factors that influence the training of Paralympic athletes and the barriers they encounter in general have been analyzed without specifically taking into account the sport practiced or the disability that the athletes had. So, future research could focus on analysing the differentiated and comparative training process between female and male Paralympic athletes, as well as to analyse the factors that in-fluence and limit this process according to the sport and/or disability presented by the athletes. |
Quality assessment was performed, but this important aspect of a systematic review is not discussed. Suggestion to integrate the quality of the studies and forthcoming findings in the discussion section. |
This issue has been addressed. For this purpose, the review carried out included only research of optimal quality. In this sense, it should be noted that all the articles included received a score equal to or greater than the cut-off point (≤70%) at the beginning of the evaluation process, which may be due to the requirement of the inclusion and exclusion criteria defined previously. In this way, the tool used to assess the quality of the documents can help to identify differences between the studies and to synthesise and interpret the main results of the studies. |
line 40-41: 2x with a disability in same sentence.
line 50: correct way of referring? (11).
line 87: no systematic review as exclusion, (leave 'no' out) otherwise you say that SR's were included.
line 93-94: this info was already provided supra. delete here.
line 96: not clear if these 3 'blocks' should reflect the 'PECO' search, and if these were combined (using 'AND' operator?)?
line 117-118: explain shortly how this tool works.
Line 121: 'was' should be 'were'
line 124: duplicates were removed. Is this mentionned in the methods section? Which software was used?
line 171: please find a different way than to use 'sporting life', maybe 'sports career'?
line 252: can you better clarify the link between economics and media attention?
line 256: That was 20 years ago. A lot of things may have changed since then. Therefore it would be interesting to see if there is some evolution in parameters, which should be possible due to no year of publication restriction in your search. Anyway, the evolution over time considering this subject should be discussed. |
The suggested changes have been made:
In the case of athletes with a disability, there is also previous research which in-dicates the existence of numerous factors influencing their training.
Corrected.
Corrected.
Corrected.
That issue has been clarified: The blocks were combined with the Bolean operator AND. PICO components (population, intervention, control group and outcome) were considered.
Explained: This tool is based on a systematic scoring system for evaluating the quality of studies and serves to ensure a minimum of quality in a systematic review [23]. For quantita-tive studies, 14 items were scored ("yes"=2, "partial"=1, "no"=0). Items not applicable to a particular study were marked 'n/a' and were excluded from the calculation of the summation score. The scores of the qualitative investigations were estimated in a sim-ilar way, considering the scores obtained in 10 items.
Corrected.
Corrected. 346 duplicate articles were removed, using excel version 2020, leaving 2046.
Corrected.
Economic barriers and media attention have been treated separately, since we have not found studies that link them.
To avoid confusion, the statement has been removed. |
THANK YOU for your comments and suggestions.

Round 2
Reviewer 3 Report
In the revised version, authors have addressed my sugestions and made some changes accordingly. However, on some aspects I feel there might be a misunderstanding, or I get the feeling that the authors didn't want to put much effort in improving their paper based on my suggestions.
For instance the aspect of the correct terminology: the term para-athletes refers to athletes with a disability, whilst the term 'paralympier' refers to a disabled athlete competing at the Olympics for people with a disability. Choosing the correct terminology is essential for the correct interpretation of this systematic review.
Secondly, the suggestion to differentiate the findings in function of the type of disability is something that effectively can be done with the results obtained. Simply giving the advice to study this in future research is a little foolish as the same work has to be re-done. I do imagine that there is probably a need for adjusted recommendations depending on the type of disability for those people that coach para-athletes.
Thirdly, the same accounts for the suggestion to take a closer look at the date of publications and reflect on the idea that there might be a certain evolution in societal norms and values that have influenced barriers or needs in this population. In 20 years, a lot can happen...
Fourthly, the link between economics and media attention is quite obvious and I am sure you will find a suitable reference that shows the link between the both (allbeit in a non-disabled population).
I do hope that the authors can acknowledge these suggestions for further improvement of their manuscript.
Author Response
COVER LETTER
Manuscript ID: ijerph-1747418. Type of manuscript: Article. Title: The sport training process of Paralympic athletes. A systematic review.
Reviewer 3’s comments and suggestions for authors |
Details of the revisions and responses |
For instance the aspect of the correct terminology: the term para-athletes refers to athletes with a disability, whilst the term 'paralympier' refers to a disabled athlete competing at the Olympics for people with a disability. Choosing the correct terminology is essential for the correct interpretation of this systematic review. |
It has been amended as suggested by the reviewer. |
Secondly, the suggestion to differentiate the findings in function of the type of disability is something that effectively can be done with the results obtained. Simply giving the advice to study this in future research is a little foolish as the same work has to be re-done. I do imagine that there is probably a need for adjusted recommendations depending on the type of disability for those people that coach para-athletes. |
It has been amended as suggested by the reviewer.
The findings of the review indicate that some disabilities have been studied more than others. In this regard, visual and physical disabilities were highlighted. In addition, there were differences in the factors and barriers studied according to the type of disability. In this way, and with regard to visual impairment, the main considerations referred to the fact that the practice of sport had a great impact on the self-confidence and self-esteem of athletes, and the need to work on the control of pressure and stress. In addition, the main barriers faced by the visually impaired were environmental, financial and poor media coverage, and their main demands were for greater social recognition, increased economic benefits, as well as better health care [16, 25-26, 28, 30-31, 34-35, 37-38]. As for para-athletes with some amputation, it should be noted that the main barriers faced by them were environmental ones, and that personal factors were their main facilitators [25, 31, 36]. Studies in para-athletes with cerebral palsy revealed the need to work on some psychological aspects such as self-confidence, motivation, and anxiety control. In addition, the main facilitators were the people around them, the desire to win a competition and the socialization provided by the practice of sport [25, 32, 39]. In people with hearing impairment, the practice of sport contributed positively to self-confidence, self-esteem and control of anxiety, while the main facilitators were related to the people around them, the desire to win a competition and socialization [30, 32]. Regarding para-athletes with some physical disability, the practice of sport was also positive. However, it is recommended to continue working on controlling the pressure exerted by sports, self-confidence, motivation, and anxiety. This question is important because psychological care in para-athletes is not usually common [38]. On the other hand, the main barriers faced by persons with physical disabilities were environmental, financial, poor media coverage and low recognition as professional athletes. The main facilitators were family, socialization, enjoyment of sports, and improved self-perception. However, these athletes demanded greater social and economic benefits [16, 28-29, 30, 33-35, 37-39]. As for people with a spinal cord or neurological injury, they considered that the main barriers they encountered were the environmental ones, while the facilitators had to do with personal factors [31]. With regard to people with intellectual disabilities, the studies revealed the need to work on the control of the pressure exerted by the practice of sports on people and sports technique [35, 40]. Finally, the study by Sánchez-Pay and Sanz-Rivas [36] showed the need to work on physical condition in athletes with osteogenesis. In summary, the above considerations should be taken into account in the training process of para-athletes in order to improve or optimize the training process. |
Thirdly, the same accounts for the suggestion to take a closer look at the date of publications and reflect on the idea that there might be a certain evolution in societal norms and values that have influenced barriers or needs in this population. In 20 years, a lot can happen... |
It has been amended as suggested by the reviewer.
In this regard, Cid [118] states that, throughout history, persons with disabilities have been subjected to rejection and discrimination and that they are usually relegated to the background in society. At present, however, progress is evident and we can see how these people are integrating into the world of work and, little by little, are overcoming some of the many barriers that affect them. She concludes by saying that it is essential to continue this work until all persons with disabilities fully enjoy their rights to integrity, education, work, non-discrimination and full integration into society. |
Fourthly, the link between economics and media attention is quite obvious and I am sure you will find a suitable reference that shows the link between the both (allbeit in a non-disabled population). |
It has been amended as suggested by the reviewer.
In this vein, Marín [119] points out that it is now clear that sporting activity, especially elite sport, has achieved great interest from the economic and commercial point of view, as well as from the media’s follow-up. This author continues to comment that the professionalization of sport has led to the commercialization of the athlete’s image through activities such as marketing, merchandising, image rights or media presence. Businesses and the media tend to focus on sports that have the greatest impact on television. |
THANK YOU for your comments and suggestions.
